# Evolution of the Bromate Electrolyte Composition in the Course of Its Electroreduction inside a Membrane–Electrode Assembly with a Proton-Exchange Membrane

**DOI:** 10.3390/ijms242015297

**Published:** 2023-10-18

**Authors:** Dmitry V. Konev, Pavel A. Zader, Mikhail A. Vorotyntsev

**Affiliations:** 1Federal Research Center for Problems of Chemical Physics and Medicinal Chemistry of the Russian Academy of Sciences, Chernogolovka 142432, Russia; 2Frumkin Institute of Physical Chemistry and Electrochemistry of the Russian Academy of Sciences, Moscow 119071, Russia

**Keywords:** Br atom-containing species, redox transitions, pH variation, potential and concentration effects, thermodynamic equilibrium, redox charge of solution

## Abstract

The passage of cathodic current through the acidized aqueous bromate solution (catholyte) leads to a negative shift of the average oxidation degree of Br atoms. It means a distribution of Br-containing species in various oxidation states between −1 and +5, which are mutually transformed via numerous protonation/deprotonation, chemical, and redox/electrochemical steps. This process is also accompanied by the change in the proton (H^+^) concentration, both due to the participation of H^+^ ions in these steps and due to the H^+^ flux through the cation-exchange membrane separating the cathodic and anodic compartments. Variations of the composition of the catholyte concentrations of all these components has been analyzed for various initial concentrations of sulfuric acid, c_A_^0^ (0.015–0.3 M), and two values of the total concentrations of Br atoms inside the system, c_tot_ (0.1 or 1.0 M of Br atoms), as functions of the average Br-atom oxidation degree, *x*, under the condition of the thermodynamic equilibrium of the above transformations. It is shown that during the exhaustion of the redox capacity of the catholyte (*x* pass from 5 to −1), the pH value passes through a maximum. Its height and the corresponding average oxidation state of bromine atoms depend on the initial bromate/acid ratio. The constructed algorithm can be used to select the initial acid content in the bromate catholyte, which is optimal from the point of view of preventing the formation of liquid bromine at the maximum content of electroactive compounds.

## 1. Introduction

Electrochemical processes with the participation of bromates as a reagent or the reaction product have been extensively studied in the literature. In particular, such transformations were carried out inside devices containing membrane–electrode assemblies or analogous arrangements where the bromide oxidation took place where bromate was a target or byproduct. Such studies were performed for bromate electrosynthesis [1,2,3], production of “active bromine” for disinfection [4,5], electrolysis of seawater for hydrogen production [6] as well as the electrochemical treatment of wastewater with the use of the BDD electrodes [7]. The inverse process of the bromate electroreduction was also studied by researchers, primarily in the context of bromate removal generated during water treatment by ozone [8]. In some of these studies [4,5], the electrolyte composition was analyzed theoretically on the basis of the equilibrium relations between various bromine-containing redox components. For reactions involving bromate interactions with reducing solutions in the course of redox titrations, a similar analysis was also performed in Refs. [9,10]. However, due to various reasons (low concentrations of bromine compounds compared to the overall salt content; addition of buffer or correcting additives to the solution), the performed analysis was limited to calculating the equilibrium composition and the redox potential of the solution at a particular pH value, or for the range of pH values determined by the acid-base equilibria between Br-containing species, taking into account the effect of a titrating agent. A similar approach was earlier used in Refs. [11,12], where the variation in the composition of a bromine-containing solution in the course of its electrolysis was analyzed.

Another area where electrochemical transformations of bromate ions turned out to be of importance is related to power sources. The transition from fossil energy sources to renewable ones is largely associated currently with hydrogen, called frequently the “fuel of the future” [13,14,15]. In turn, its generation and use are most efficiently carried out by electrochemical devices, i.e., electrolyzers and power sources [16,17]. Among the latter, fuel cells (FCs) based on the use of atmospheric oxygen as the oxidizing agent are currently the most developed ones [18]. Despite the considerable maturity of the technology of hydrogen–air FCs and the variety of these devices [19,20], their ubiquitous distribution is largely constrained by issues arising from the specifics of the electrode reaction of the oxygen half-cell: the need to use large amounts of expensive catalysts, sensitivity to catalyst poisoning, and aggressive intermediates of the oxygen electroreduction process. These issues have stimulated further search for new types of power sources with the use of hydrogen fuel [21,22]. One such device is a hydrogen-bromate battery, which represents the combination of a hydrogen gas-diffusion anode with a cathode where the acidized aqueous solution of a salt of the bromic acid passes through the porous carbon material [23,24,25,26,27]. During the discharge of the device, the bromate anion is reduced by hydrogen to the corresponding bromide according to the global equation:3H_2_ + BrO_3_^−^ = Br^−^ + 3H_2_O,(1)

Due to the excellent solubility of bromates, their high redox potential, and the six-electron reduction reaction to bromide, the theoretical stored energy density (when using lithium bromate and a modern hydrogen storage system) can reach 1100 W-h/kg [28].

Previously it was shown [29] that the bromate reduction process requires a sufficiently high acidity of the medium to ensure large values of the generated current. In addition, the passage of the bromate cathode half-reaction
BrO_3_^−^ + 6H^+^ + 6e^−^ = Br^−^ + 3H_2_O,(2)
leads to the consumption of one proton per transferred electron, but this process is exactly compensated by the transfer of hydrogen ions, generated at the gas-diffusion anode, across the proton-exchange membrane into the bromate solution.

Since bromate anion does not react at any electrode surface, its sufficiently rapid transformation may only be achieved owing to the autocatalytic mediator cycle, based on the Br_2_/Br^−^ redox couple, which consists of the bromine/bromide electrode reaction and the bulk chemical stage of the bromate reduction by bromide anion, with the formation of bromine:Br_2_ + 2e^−^ = 2Br^−^,(3)
5Br^−^ + BrO_3_^−^ + 6H^+^ = 3Br_2_ + 3H_2_O,(4)

The rate of this process increases rapidly for a higher acidity of the bromate solution. However, the use of acidified concentrated bromate catholytes results in the accumulation of a high concentration of dissolved molecular bromine that exceeds its solubility (saturation) limit at the medium stage of the whole process, thus leading to liquid bromine formation. The latter disrupts the operation of the cell due to corrosiveness, volatility, and low electrical conductivity.

The current study has been devoted to a possible way to resolve this problem, i.e., to diminish the amount of molecular bromine, Br_2_, which is accumulated inside the catholyte in the course of the bromate, BrO_3_^−^, electroreduction. The principal idea has been to invert the ratio of the starting bromate and acid concentrations in the starting solution, namely, to take a ***lower acid concentration***. One has to keep in mind that such a variation in the initial composition means inevitably an ***even stronger diminution of the proton concentration in the course of the electrolysis***.

Therefore, the goal of this study has been to carry out a theoretical analysis of the evolution of the composition of the acidized bromate solution in the course of its electroreductive transformation into the bromide one, taking into account the variation in the concentrations of ***all components of the system including that of protons***. Such a consideration has been performed on the basis of thermodynamic ***relations between the concentrations of the solute components***.

Thus, the essential difference of this work from our earlier analysis [11,12] is the accounting for the change in the concentration of hydrogen ions inside the catholyte due to the balance between their generation at the anode (hydrogen oxidation) and consumption within the cathodic compartment due to the progressive transformation of oxygen-containing forms of bromine (BrO_3_^−^, BrO^−^, HBrO) into oxygen-free species (molecular bromine, bromide anion, and complexes between them) coupled with the formation of water.

## 2. Results and Discussion

### 2.1. Theoretical Analysis

Let us assume that the electrolysis of the Br-containing solution is carried out under conditions where the electrochemical charge, *Q*_elchem_, passed by a time moment of the electrolysis process, *t*, is fully spent for the change in the oxidation degrees of Br atoms inside the system. It gives Equation (5):*Q*_elchem_ = *Q* − *Q*_ini_,(5)
i.e., the measured value of *Q*_elchem_ provides direct information on the difference between the total redox charges of the system prior to electrolysis, *Q*_ini_, and at the chosen time moment, *Q* (see its definition in Section 2.1.1 below):

Since the initial value of the total redox charge, *Q*_ini_, can be calculated on the basis of the procedure of the starting solution preparation, Equation (5) enables one to monitor the variation in the total redox charge of the system, *Q*, as a function of time, *t*.

#### 2.1.1. Definitions and Balance Relations

Analogously to Refs [11,12], we take into account below the stable solute Br-containing species of oxidation degrees from −1 to +5 inside the solution phase; see species from i = 1 to i = 7 in Table 1.

Transitions into the highest oxidation state, +7, are not considered because the standard potential of its formation (over 1.8 V) is far outside the potential range where the evolution of the potential takes place in the course of the reductive bromate-anion (BrO_3_^−^) electrolysis.

Non-dissociated forms of strong acids, HBr, HBr_3_, HBr_5_, and HBrO_3_, are not considered since their concentrations with respect to the corresponding ionic ones are very small, at least within the whole range: pH > 0; e.g., the signal of HBrO_3_ is not visible in Raman spectra of 3 M HBrO_3_ [30].

Under conventional experimental conditions, the volume of the gas space above the solution is small with respect to or comparable with the volume of the solution. Then, one may neglect in the balance relations below the contribution of ***bromine molecules inside the gas phase***, Br_2_^vap^ (i = 9 in Table 1) since even under equilibrium for the exchange by bromine molecules between the solution and the gas phase the amount of the former component is much larger than the latter one because of a relatively low evaporation constant [11,12].

Generally, because of the relatively low solubility of Br_2_ in water, one has to take into account the possibility of the formation of the liquid bromine phase, Br_2_^liq^ (i = 8 in Table 1), which is in equilibrium with solute Br_2_ at its saturated concentration: [Br_2_,_sat_] = 0.185 M [31]. The method to calculate the evolution of the solution composition under the condition where the liquid bromine phase ***may*** be formed was described in Refs [11,12]. The goal of this study has been to determine the conditions, i.e., the interval of the added acid concentrations, where the liquid bromine phase ***is not formed***. Therefore, the possible contribution of the liquid bromine to all balance relations will be ***disregarded*** in the analysis below. Then, ***if*** the solute bromine concentration in the course of the electrolysis remains ***always below its saturation limit***: [Br_2_] ≤ [Br_2_,_sat_]_,_ such initial values of the parameters of the system are ***favorable for the electrolysis***. On the contrary, if this concentration exceeds the saturation limit within the medium time range, such initial conditions are ***inappropriate for it***.

Thus, only the ***solute*** Br-containing components given by columns for i = 1 to i = 7 are taken into account for the analysis below.

Table 1 also contains a row of the parameters, *n*_i_, which are equal to the number of Br atoms inside ***one*** species of the corresponding type, i.

Another useful parameter of each species is its ***total oxidation number*** of Br atom(s) inside ***one*** species of this type, *x*_i_. It is defined with the use of the corresponding electrochemical scheme where one species of this type, i, is transformed into solute neutral bromine Br_2_ molecule (i = 4), for which the total oxidation number is taken as zero: *x*_4_ = 0, e.g.,
Br^−^ − e^−^ = ½Br_2_, Br_3_^−^ − e^−^ = 3/2Br_2_, BrO_3_^−^ + 6H^+^ + 5e^−^ = ½Br_2_ + 3H_2_O,(6)
i.e., one should withdraw one electron from one species of type 1 (i = 1), Br^−^, so that *x*_1_ = −1, or withdraw one electron from one species of type 2 (i = 2), Br_3_^−^, so that *x*_2_ = −1, or add five electrons to one species of type 7 (i = 7), BrO_3_^−^, so that *x*_7_ = +5. For species containing only one Br atom, the value of the thus defined parameter, *x*_i_, is equal to the conventional ***oxidation degree*** (***oxidation number***) of its Br atom while it avoids fractional numbers for species including several Br atoms; see Table 1.

If the Br-containing species do not leave the working electrode compartment in the course of electrolysis, then the total number of moles of Br atoms ***in all Br-containing components of the system***, *N*_tot_, does not change in time so that it may be calculated on the basis of the solution preparation procedure. This value is included in the balance equation for Br atoms:*N*_tot_ = ∑ *n*_i_ *N*_i_,(7)
where the summation is carried out over all Br-containing components in Table 1 for 1 ≤ i ≤ 7, while *N*_i_ is the number of moles of Br-containing species of type i at a time moment, *t*, in the course of electrolysis; numbers *n*_i_ are given in Table 1.

The ***total redox charge*** of the Br-containing system (where all other elements including protons, H^+^, are non-electroactive, i.e., they do not change their oxidation degrees) is defined by the formula:*Q* = *F* ∑ *x*_i_ *N*_i_,(8)
where total oxidation numbers, *x*_i_, for all species may be found in Table 1, and *F*—Faraday constant; see also Equation (5).

***The average oxidation degree***, *x*, of Br atoms in this state of the system characterizes the average redox charge per one Br atom (in dimensionless units) of the whole system:*x* = *Q/F N*_tot_ = ∑ *x*_i_ *N*_i_/∑ *n*_i_ *N*_i_,(9)

By definition, the value of *x* is equal to 0 for the solution obtained by dissolution of pure bromine, Br_2_ (even though this species may be subjected to various transformations, e.g., to hydrolysis) in water while, according to Equation (9), it varies from −1 for pure Br^−^ solution to +5 for pure BrO_3_^−^ one.

If the system in its initial state (prior to electrolysis) is prepared by dissolution of *N*_ini_ moles of a substance where each species (molecule or ion) contains *n*_ini_ Br atoms and where the total oxidation number is equal to *x*_ini_, then *N*_tot_ = *n*_ini_ *N*_ini_, and *Q*_ini_ = *F x*_ini_ *N*_ini_. For example, *n*_ini_ = 1, *x*_ini_ = −1 for the initial NaBr solution, and *n*_ini_ = 2, *x*_ini_ = 0 for the initial Br_2_ solution while one has *n*_ini_ = 1, *x*_ini_ = +5 for the initial NaBrO_3_ solution. If the initial solution is prepared by dissolution of two Br-containing substances, the values of *N*_tot_ and *Q*_ini_ are given by the sum of their contributions. The presence of other solute species that are non-electroactive (i.e., do not participate in redox reactions at the electrode or inside the solution) does not affect these relations.

The knowledge of values of *Q*_ini_ and of experimentally measured electrolysis charge, *Q*_elchem_, at some time moment, *t*, enables one to calculate the total redox charge of the system for this moment, *Q* = *Q*_ini_ + *Q*_elchem_, Equation (5).

Since the total number of Br atoms in all components of the system does not change in the course of the electrolysis, Equation (7) may be rewritten in terms of the concentrations of the components, *c*_i_:*c*_tot_ = ∑ *n*_i_ *c*_i_,    where   *c*_tot_ = *N*_tot_/*V*,   *c*_i_ = *N*_i_/*V*,(10)

*V* is the volume of the solution. ***The total Br-atom concentration***, *c*_tot_, does not vary in time. Equations (8) and (9) take the form:*Q* = *F V c*_Q_,  *x* = *c*_Q_/*c*_tot,_     where *c*_Q_ = ∑ *x*_i_ *c*_i_,(11)

Our previous analyses in Refs [11,12] were carried out for the system ***containing a pH buffer*** in the solution so that the pH value of the solution ***remained constant*** throughout the electrolysis, despite the passage of the proton-containing steps. On the contrary, it is assumed below that prior to the electrolysis, the solution contains both a Br-containing species (or their mixture) and an added acid as the source of protons, H^+^.

The further consideration below is performed for the initial composition of the solution which contains bromate anion, BrO_3_^−^ (concentration: *c*_tot_) and sulfuric acid (concentration: *c*_A_). Then, the initial redox charge of the system is given by the formula:*Q*_ini_ = 5 *FV c*_tot_,(12)

Balance relations for H and O atoms in the system may be written as the equality of each quantity for the initial state (prior to electrolysis) and for any moment in the course of the electrolysis:*N*_H_^0^ + *N*_HA_^0^ + *N*_HQ_ = *N*_H_ + *N*_HA_ + 2*N*_H2O_ + ∑ *h*_i_ *N*_i_,    3*N*_tot_ = *N*_H2O_ + ∑ *o*_i_ *N*_i_,(13)
where, for the initial state and for the chosen moment in the course of the electrolysis, respectively, *N*_H_^0^ and *N*_H_ are the numbers of solute protons (H^+^), *N*_HA_^0^ and *N*_HA_ are the numbers of bound protons left inside non-dissociated acid (H_2_SO_4_) or inside HSO_4_^−^, and *N*_HQ_ is the number of protons transferred across the separating membrane as a result of the passage of the electrolysis charge, *Q*_elchem_, so that *N*_HQ_ = (*Q*_ini_ − *Q*)/*F,* according to Equation (5). The reduction process is accompanied by the formation of *N*_H2O_ water molecules as well as a partial transformation of BrO_3_^−^ ions into other Br-containing components (1 ≤ i ≤ 7 in Table 1), some of them containing H or/and O atoms (their numbers in one species of the corresponding component, *h*_i_ and *o*_i_, are also listed in Table 1). It is also assumed that the solution during the electrolysis retains its ***acidic*** (pH < 7) character, in conformity with the calculations below.

One should keep in mind that the total number of free and bound protons in the initial sulfuric acid solution (even in the presence of a neutral bromate salt), *N*_H_^0^ + *N*_HA_^0^, is equal to 2 *V* c*_A_*.

The combination of two Equations (13) gives a relation for the concentrations of the Br-containing components (*c*_i_) as well as of H^+^, HSO_4_^−^, and H_2_SO_4_ (*c*_H_, *c*_HSO4,_ and *c*_H2SO4_) in the initial and intermediate states:2 *c*_A_ = *c*_H_ + *c*_HSO4_ + 2 *c*_H2SO4_ + *c*_tot_ (1 + *x*) + ∑ *h*_i_ *c*_i_ − 2 ∑ *o*_i_ *c*_i_,
or 2 *c*_A_ = *c*_H_ + *c*_HSO4_ + 2 *c*_H2SO4_ + *c*_tot_ + ∑ *x*_i_ *c*_i_ + ∑ *h*_i_ *c*_i_ − 2 ∑ *o*_i_ *c*_i_,(14)

The use of parameters for the components in Table 1 allows one to write down Equations (10) and (11) in the explicit form:*c*_tot_ = [Br^−^] + 3[Br_3_^−^] + 5[Br_5_^−^] + 2[Br_2_] + [HBrO] + [BrO^−^] + [BrO_3_^−^],(15)
*x c*_tot_ = *Q*/(*F V*) = −[Br^−^] − [Br_3_^−^] − [Br_5_^−^] + [HBrO] + [BrO^−^] + 5[BrO_3_^−^],(16)

Equilibrium conditions for the two-step dissociation of sulfuric acid give two relations between three concentrations, *c*_H_, *c*_HSO4,_ and *c*_H2SO4_ for any state of the system. The addition of the balance relation for these components allows one to express two concentrations, *c*_HSO4_ and *c*_H2SO4_, as functions of the proton concentration, *c*_H_.

The value of the dissociation constant of sulfuric acid, *K*_a_, is well established for the ***second*** step, HSO_4_^−^ = H^+^ + SO_4_^2−^: pK_a_ = 1.99, i.e., *K*_a_ ≅ 0.01 [32].

The dissociation constant for the ***first*** step, H_2_SO_4_ = H^+^ + HSO_4_^−^, is much larger: its pK_a_ value is negative [32]. To avoid the use of inaccurate values of this constant, it is assumed below that one can neglect the presence of ***non-dissociated*** acid molecules, H_2_SO_4_, in solution for its concentration range well below 1 M. It gives approximate relations for any moment of the electrolysis process:*K_a_* = *c_H_ c_SO4_*/*c_HSO4_*,   *c_SO4_* + *c_HSO4_* ≅ *c_A_*, i.e.,  *c_HSO4_* ≅ *c_A_ c_H_* (*K_a_* + *c_H_*)^−1^,   *c_SO4_* ≅ *c_A_ K_a_* (*K_a_* + *c_H_*)^−1^,(17)

As a result, Equation (14) can be written down with the use of Equations (17) and (18) in the form:2 *c*_A_ ≅ *c*_H_ + *c*_HSO4_ + *c*_tot_ (1 + *x*) − [HBrO] – 2[BrO^−^] – 6[BrO_3_^−^],or*c*_H(H2SO4)_ ≅ *c*_H_ + *c*_tot_ − [Br^−^] − [Br_3_^−^] − [Br_5_^−^] – [BrO^−^] – [BrO_3_^−^],(18)
where the bisulfate concentration at any stage of the electrolysis, *c*_HSO4_, is a function of the proton concentration, *c*_H_, of this solution, Equation (17), while
*c*_H(H2SO4)_ ≅ 2 *c*_A_ − *c*_HSO4_ = *c*_A_ (2 *K*_a_ + *c*_H_) (*K*_a_ + *c*_H_)^−1^,(19)
is the concentration of ***solute*** protons, [H^+^], generated by sulfuric acid if the solution has finally achieved the *c*_H_ concentration of such species.

The total dissociation degree of the sulfuric acid, α_H_, which is defined here as the number of solute protons produced by one added acid molecule may be expressed via the added acid concentration, *c*_A_, and the concentrations of the solute components, *c*_HSO4_ and *c*_SO4_ (the presence of the solute non-dissociated acid, H_2_SO_4_, is disregarded, see above) with the use of the balance relations and of approximate Equation (17):*c*_H(H2SO4)_ ≅ 2 *c*_A_ − *c*_HSO4_ = *c*_A_ (2 *K*_a_ + *c*_H_) (*K*_a_ + *c*_H_)^−1^,(20)
where *c*_H_ is the solute proton concentration due to all its sources, Equations (14) or (18).

#### 2.1.2. Thermodynamic Relations

Within the potential range of this study, only Br atoms inside solution components change their oxidation degrees (between −1 and 5). Under equilibrium conditions, the activities of each two of the Br-containing species are linked by the corresponding Nernst relation (for different oxidation degrees of these species) or by the physicochemical one (e.g., for the acidic dissociation); see Table A1 in Appendix A. For the set of seven Br-containing components of the system (1 ≤ i ≤ 7 in Table 1) it is enough to use six independent relations, e.g., those that link each of the six components of higher oxidation degrees (2 ≤ i ≤ 7) with bromide anion (i = 1).

Similar relations take place for the concentrations of these species, with a shift of the thermodynamic parameters due to activity coefficients [33]. For the relatively dilute solutions under consideration, these changes are small, compared to a much larger scale of variation in the parameters, in particular, of the electrode potential, E, and pH of the solution, see below. Therefore, this effect will be disregarded.

#### 2.1.3. Set of Equations

There are two choices of the independent electric variable: electrode potential, *E*, or total redox charge of the system, *Q*. The calculation algorithm in this study uses the former variant. Then, after determination of the evolution of the solution composition, one can calculate the variation in the redox charge, *Q*, as well as of the average Br-atom oxidation degree, *x*, in the course of the process owing to Equation (16).

In view of the thermodynamic basis of the analysis, the alternative choice of the redox charge, *Q* (or *x*), as the independent variable would result in an exactly identical relation between the potential and the redox charge/average oxidation degree.

As demonstrated in Section 2.2, Section 2.3, Section 2.4 and Section 2.5, both the electrode potential, *E*, and the redox charge, *Q*/average Br-atoms oxidation degree, *x*, vary ***monotonously in the negative direction*** in the course of the electrolysis, thus resulting in the evolution of the solution composition, with progressive transition from bromate anion (*x* = +5) to bromide anion (*x* = −1) via various intermediate Br-containing components.

Unlike our previous analysis of this process in the presence of a pH buffer that fixes a certain pH value of solution, the concentration of solute protons, [H^+^] = *c*_H_, ***varies*** in the course of the evolution, its variation being ***non-monotonous***. The latter conclusion can be drawn from Equation (18): both for the initial and the final states where the solution contains only bromate (*x* = +5), or only bromide (*x* = −1) anion, respectively, the solute proton concentration, *c*_H_, satisfies the same equation: 2 *c*_A_ ≅ *c*_H_ + *c*_HSO4_; i.e., ***pH returns to the same initial value at the end of the evolution*** while its values are ***different*** for the intermediate region of the electrode potential, *E*, or of the redox charge, *Q*/average Br-atoms oxidation degree, *x*.

For the analysis of the electrolysis process in this system, one should solve a set of equations: Equations (15) and (18) as well as (A1)–(A3), (A5), (A6), (A10) in Table A1, for seven concentrations of Br-containing compounds (1 ≤ i ≤ 7 in Table 1) and the solute proton concentration (*c*_H_), for various values of the electrode potential, *E*. The total concentration of Br atoms inside the system, *c*_tot_, keeps a constant value. The indicated relations in Table A1 allow us to express each of the six concentrations of Br-containing species (2 ≤ i ≤ 7) via the bromide-anion concentration, [Br^−^], and the solute proton concentration, *c*_H_, i.e., solution pH, for each value of the electrode potential, *E*, in order to insert them into Equations (15) and (18). It gives a set of two coupled algebraic equations for [Br^−^] and *c*_H_:[Br^−^] = *c*_tot_ − 3[Br_3_^−^] − 5[Br_5_^−^] − 2[Br_2_] − [HBrO] − [BrO^−^] − [BrO_3_^−^],and*c*_H_ ≅ *c*_H(H2SO4)_ − *c*_tot_ + [Br^−^] + [Br_3_^−^] + [Br_5_^−^] + [BrO^−^] + [BrO_3_^−^],(21)

### 2.2. General Shapes of Various Dependencies

Here, shapes of various numerically calculated dependences are discussed for the total Br-atom concentration (equal to the initial concentration of bromate anion, [BrO_3_^−^]), *c*_tot_ = 0.1 M. As for the added concentration of sulfuric acid, two different values are considered in this section as illustrations (more extended ranges of *c*_A_ and *c*_tot_ are discussed in Section 2.4 and Section 2.5): *c*_A_ = 0.3 M and *c*_A_ = 0.03 M, which illustrate features of the evolution for the excess of protons (compared to that of bromate anions) or of bromate anions (compared to that of protons), respectively.

Figure 1 illustrates relations between the global characteristics of the system in the course of the electrolysis process: electrode potential, *E*, average oxidation degree of Br atoms, *x*, proportional to the total redox charge of the system, *x* = *Q*/*FV c*_tot_, and pH of the solution.

Dependencies of *x* vs. *E* are given in Figure 1a. The shape of the plot ***for a relatively high acidity, c_A_ =* 0.3 *M***, pH ≅ 0.52 (red line) resembles that for *c*_tot_ = 0.1 M and ***fixed*** pH = 2 (black line 2 in Figure 2 of [12]), with a shift of the first reduction wave (BrO_3_^−^/Br_2_ transition, 0 < *x* < +5) for the plot in Figure 1a towards higher potentials, due to lower pH values, and the same position of the second reduction wave (Br_2_/Br^−^ transition, −1 < *x* < 0).

This observation corresponds to the expectations. Even though the proton concentration is ***variable*** in Figure 1, the first dissociation degree of H_2_SO_4_ gives already a much larger number of solute protons than that of bromate anions. The loss of one equivalent of BrO_3_^−^ according to Scheme (22):BrO_3_^−^ + 6H^+^ + 5e^−^ = ½Br_2_ + 3H_2_O,(22)
is accompanied by the consumption of six equivalents of H^+^, which is partially compensated by the transfer of 5 H^+^ across the membrane, i.e., the total transformation of 0.1 M BrO_3_^−^ does not consume more than 0.1 M protons so that the acidity of solution does not change drastically in the course of the electrolysis (red lines for pH variation in Figure 1b,c).

Figure 1b,c illustrate the dependence of pH on electrode potential, *E*, or on the average oxidation degree of Br atoms, *x* = *Q*/*FV c*_tot_. In both cases the pH value for *c*_tot_ = 0.1 M, *c*_A_ = 0.3 M (red lines) varies within a relatively narrow range, from its initial value close to –lg *c*_A_ ≅ 0.52 up to its maximum around −lg (*c*_A_ − *c*_tot_) ≅ 0.70, with its subsequent return finally to the initial value.

These conclusions for *c*_tot_ = 0.1 M, *c*_A_ = 0.3 M are supported by the data in Figure 2a,b on the evolution of the solution composition in the course of the electrolysis.

According to Figure 2a, for the dependence of the solute proton concentration, *c*_H_, on electrode potential, *E*, the concentration (black line) varies from about 0.31 M (slightly above 0.3 M due to the second dissociation degree of the acid) to its minimal value around 0.21, with its return to the initial value at the end of the evolution. Variation in the same concentration as a function of the average oxidation degree of Br atoms, *x* = *Q*/*FV c*_tot_ (black line in Figure 2b), shows a similar tendency while its shape is markedly different, being close to two straight lines between its maximal and minimal values.

Red lines in Figure 2a,b illustrate the variation in the bromate ion concentration, [BrO_3_^−^] for *c*_tot_ = 0.1 M, *c*_A_ = 0.3 M, with its practically full transformation to Br_2_, scheme (22). Unlike a typical Nernstian wave for the dependence: [BrO_3_^−^] vs. *E*, with a very narrow transition range because of the five-electron process (Figure 2a), Equation (22), the dependence of this concentration on the average oxidation degree of Br atoms, *x* = *Q*/*FV c*_tot_ (Figure 2b), is close to a straight line, where the concentration varies from its initial value, 0.1 M, to a very small one at *x* = 0.

According to Figure 2a, the evolution of Br_2_ concentration vs. potential at *c*_tot_ = 0.1 M, *c*_A_ = 0.3 M (blue line) includes a drastic increase from 0 to its maximal value (close to one-half of 0.1 M, Scheme (22)), followed by an almost constant (slightly decreasing) value, up to the second (downward) Nernstian wave. In the coordinates: [Br_2_] vs. *x* = *Q*/*FV c*_tot_ (blue line in Figure 2b), the concentration varies between the same minimal and maximal values, but the shape of this dependence is quite different, being close again to a straight line within the region: 0 < *x* < +5, Scheme (22)) while there is a marked deviation from linearity within the downward region (−1 < *x* < 0), which reflects obviously the formation of a significant amount of intermediate species, Br_3_^−^ and Br_5_^−^. This explanation is confirmed by magenta and green lines in Figure 2a,b for the Br^−^ and Br_3_^−^ concentrations, respectively, where the number of Br atoms inside Br_3_^−^, *n*_i_ *c*_i_ for i = 2 in Table 1, is comparable with those for Br^−^ and BrO_3_^−^ in the middle of the transitional region.

All these qualitative results are in full conformity with those found earlier (Figure 3a in [12] and Figure 2.1 for pH = 2 in [33]), except for the absence of the liquid bromine phase due to a lower amount of Br atoms in the system under consideration, *c*_tot_ = 0.1 M.

Let us discuss general shapes of the same dependences for ***a lower acidity***, i.e., for a ***lower concentration of the added sulfuric acid***: c_tot_ = 0.1 M, ***c_A_ =* 0.03 M** (black lines in Figure 1 and Figure 2c,d).

Comparison of red and black lines in Figure 1 demonstrates immediately a radical difference in their shapes for *c*_A_ = 0.3 M and *c*_A_ = 0.03 M.

In particular, Figure 1a shows that a very broad plateau (x ≅ 0) at the plot of x = Q/FV *c*_tot_ vs. *E* within the medium potential range for *c*_A_ = 0.3 M (red line) has disappeared at the plot for *c*_A_ = 0.03 M (black line) where a narrow “inclined plateau” is visible at *x* ≅ +2, which separates two regions of a rapid variation in *x*. The second of these regions terminates abruptly at a potential value where it approaches the red line having a different angle but, instead of crossing the red line, the black line merges with it at lower values of *x*.

A similarly strong difference in the shapes of the red and black lines may be seen in Figure 1b,c for the pH dependence on *E* or on *x* = *Q/FVc*_tot_. While the variation in pH at the former plots is relatively small, this parameter ***changes by several units*** for the black lines, first increasing from the initial value (about 1.4) up to a maximum (about 4.6 at *E* ≅ 1.13 V or at *x* ≅ −0.2), to decline back to the initial value at the end of the process. One may note the existence of two regions of the almost linear dependence for the black line in Figure 1b: close to a constant value for *E* > 1.38 V or to an inclined straight line within the intermediate range: 1.13 V < *E* < 1.37 V), as well as a drastic drop within a very narrow range of *E* (or range of *x* in Figure 1c). The lines in the coordinates: pH vs. *x* (Figure 1c) reflect the same features, but their shape is deformed due to the nonlinear relation between the potential, *E*, and the average oxidation charge, *x* (Figure 1a).

Keys for interpretation of such a behavior of the principal parameters of the system may be found in the data for the evolution of the solution composition (Figure 2c,d).

As one can see from the black lines in Figure 2c,d, the proton concentration, c_H_, diminishes in the course of the electrolysis from its initial value, *c*_H_ = 0.036 M (its difference from the concentration of the added acid, *c*_A_ = 0.03 M, is obviously due to the second dissociation degree of the acid), up to very low values within the intermediate range of the potential, *E*, or of the average oxidation degree of Br atoms, *x = Q/FVc*_tot_, which correspond to high values of pH in Figure 1b,c. The c_H_ concentration returns to its initial value after a rapid increase within the range: *E* < 1.13 V or *x* < −0.4.

This behavior may be interpreted via Scheme (22), where 5 of 6 H^+^ ions are compensated by the transmembrane transport. It predicts a linear variation in both concentrations, c_H_ and [BrO_3_^−^], as functions of *x*, with the same slope. This prediction is confirmed by data in Figure 2d for the starting range of the process: +3.5 < *x* < +5 for c_H_ and +2 < *x* < +5 for [BrO_3_^−^]. The former interval is narrower, with a slower decrease in the c_H_ concentration (Figure 2d) within the range: +2 < *x* < +3.5. It is shown below (Section 2.3) that this deviation from the linearity is due to a variable contribution of the second dissociation step of sulfuric acid.

Approximately linear decrease (with a bit smaller slope) of the bromate ion concentration, [BrO_3_^−^], continues within the next range: *x* < +2 (black line in Figure 2d), with the concentration approaching zero at *x* ≅ −0.4.

As one can see from Figure 2d for the Br_2_ concentrations as a function of x, its value varies linearly within the range: +2 < *x* < +5, then it changes in the opposite direction (nonlinearly) in the interval: −0.7 < *x* < +2, with an abrupt increase in the slope at *x* ≅ −0.7. The dependences of the Br^−^ and Br_3_^−^ concentrations (Figure 2d) show a complimentary behavior: very small values within the range: +2 < *x* < +5, then a linear (for Br^−^) or nonlinear (for Br_3_^−^) increase in the interval: −0.7 < *x* < +2, with an abrupt increase (for Br^−^) or decrease (for Br_3_^−^) of the slope in the vicinity of *x* ≅ −0.7.

Comparison of data in Figure 2d for the variation in various concentrations with the pH dependence in Figure 1c allows us to draw the conclusion that the range of x values: −0.4 < *x* < +5 where c_H_ decreases monotonously coincides with the range where BrO_3_^−^ is transformed first into Br_2_ and later directly into Br^−^.

The same conclusion may be made on the basis of a comparison between Figure 2c and Figure 1b for the dependences of various concentrations on the electrode potential, *E*. However, the shapes of these plots are more complicated. In particular, unlike the plots vs. *x*, where the transitions between different ranges take place at certain points (*x* = +3/5, or +2, or −0.4), the transitions in the plots vs. *E* occur within extended regions, i.e., without abrupt changes of the slope. Another characteristic feature of the latter plots is the absence of plateau-like regions.

In conformity with these reasonings, one can notice in Figure 2c,d the practical coincidence of the points where the BrO_3_^−^ concentration seems to vanish while the proton one starts its rapid increase, as well as a similar feature of the decreasing proton concentration with increasing Br_3_^−^ and Br^−^ concentrations; the plot for the Br_2_ concentration changes abruptly its slope in the same two points.

### 2.3. Variation in Dissociation Degree of Sulfuric Acid

Since the dissociation degree of the first step of this acid is practically complete under the conditions of this study, the total dissociation degree, α_H_, should belong to the interval between 1 and 2. It is determined by the pH of the solution, i.e., by the solute proton concentration, *c*_H_, Equation (20).

In order to illustrate the variation in the dissociation degree for the second step, Figure 3 presents the dependence of the ***total*** dissociation degree of sulfuric acid, α_H_, defined by Equation (20), in the course of the reductive electrolysis of bromate, as a function of the electrode potential, *E* (Figure 3a) or of the average oxidation degree of Br atoms, *x* = *Q*/*FVc*_tot_ (Figure 3b), for the same two initial solution compositions: *c*_tot_ = 0.1 M, *c*_A_ = 0.3 M (red lines) or *c*_A_ = 0.03 M (black lines).

In the former case, the acid concentration is in excess, compared to the initial bromate concentration, so that *c*_H_ – [BrO_3_^−^] is close to 0.2 M within the whole electrolysis period (Figure 2a,b), i.e., *c*_H_ is much larger than the dissociation constant of the second dissociation stage, *K*_a_, so that Equation (20) gives: α_H_ ≅ 1 within the whole potential range, in conformity with the red lines in Figure 3a,b.

Therefore, the calculations of the other characteristics for this case might be approximately calculated within the framework of this approximate equality. This conclusion is illustrated by the proximity of the dependence of the average oxidation degree of Br atoms, *x* = *Q*/*FV c*_tot_, or of pH on the potential, *E* (Figure 3c,d), or pH on *x* (Figure 3e) for the higher concentration of sulfuric acid, *c*_A_ = 0.3 M, to the approximate line calculated for α_H_ = 1, i.e., for a strong monovalent acid.

Quite different results are obtained for the second system (*c*_tot_ = 0.1 M, *c*_A_ = 0.03 M; black lines in Figure 3a,b,e,f), where the initial concentration of solute protons satisfies the inequalities: *c*_H_^ini^ ≤ 2 *c*_A_ < *c*_tot_. In this case, the total dissociation degree of sulfuric acid, α_H_, demonstrates a strong variation as a function of *E* (Figure 3a) or of *x* (Figure 3b). Its initial value is about 1.2, which is markedly different from its both lower- and upper-limiting values equal to 1 and 2, respectively, since the initial proton concentration, *c*_H_^ini^ = α_H_ *c*_A_ ≅ 0.036, is comparable with the dissociation constant of the second dissociation step of sulfuric acid, *K*_a_ = 0.01. Then, α_H_ increases for smaller values of *E* or of *x* up to its maximal value which is close to 2 because the solute proton concentration becomes much smaller than *K*_a_ within an extended range of *E* or *x* values. Then, at a critical point, the slope of the α_H_ dependence changes abruptly and it decreases rapidly up to the value equal to the initial one, 1.2, since the solute proton concentration returns finally back to *c*_H_^ini^.

This variation in the total dissociation degree, α_H_, manifests itself in the shape of all characteristics, in particular of the average oxidation degree of Br atoms, *x* = *Q*/*FV c*_tot_, as a function of the potential, *E* (Figure 3c). In the course of the *x* diminution from its initial value, +5, the solid black line for the sulfuric acid changes between the approximate ones within the range: +2 < *x* < +5, with an approach to the line for α_H_ = 2 for the region: *x* < +2 where the solute proton concentration becomes much smaller than the dissociation constant of the second step of sulfuric acid. All three lines for this added acid concentration, *c*_A_ = 0.03 M, demonstrate the same specific feature: an abrupt change in the slope at the point ***where the bromate concentration vanishes***; see Figure 3g below. After the passage of this point, each line in Figure 3c acquires a ***universal*** behavior, being independent of the dissociation degree, or of the added acid concentration: *c*_A_ = 0.3 M or *c*_A_ = 0.03 M. This property is obviously a consequence of the absence of bromate anion or other Br-containing species of positive oxidation degrees so that the evolution is due to transitions between oxygen-free components (mostly: Br_2_, Br_3_^−^ and Br^−^), which are pH independent.

The above variation in the total dissociation degree, α_H_, also influences the shape of the evolution of the concentrations, e.g., those of the solute protons, i.e., pH of the solution, in Figure 3f,g. At the starting period of the electrolysis, the plots for the dependences of pH on *E* or on *x* are not especially far from those for the single-proton dissociation, α_H_ = 1, since the initial value of α_H_ is about 1.2. However, the further evolution results in a drastic decrease in the proton concentration, thus leading to the approach of α_H_ to its low-concentration limit, α_H_ = 2 (black lines in Figure 3a,b), so that the solid black lines for sulfuric acid approach the corresponding dashed lines for α_H_ = 2 (Figure 3f,g). This increase in α_H_ occupies an extended interval, e.g., from +5 to +2, as a function of *x*. Within the further evolution stage, the pH variation is close to a linear dependence on *E* within a broad potential range, up to its maximal value, being practically independent of the dissociation degree (Figure 3f) while the plot for pH vs. *x* for sulfuric acid approaches its approximation for α_H_ = 2 only for very low proton concentrations (Figure 3g). Finally, after a drastic drop in pH, its values approach gradually its limiting behavior for α_H_ ≅ 1.2.

Figure 3h,i illustrate the effect of the variation in the dissociation degree of sulfuric acid on the concentrations’ evolution as functions on *E* (Figure 3h) or *x* (Figure 3i) for the more interesting case of the low added-acid concentration: *c*_tot_ = 0.1 M, *c*_A_ = 0.03 M. In conformity with the above discussion of pH plots, the lines for solute proton concentration start between two approximations for α_H_ = 1 and for α_H_ = 2. Further decrease in the coordinates: *c*_H_ vs. *E* gives a universal curve (independent of the dissociation degree) while within the region of its backward increase, the plot for sulfuric acid is located between two limiting curves (Figure 3h). Unlike smooth plots in the previous case, the dependence: *c*_H_ vs. *x* reveals ***straight lines for both approximations*** for the regions of the concentration decrease and its increase (dashed and dotted lines in Figure 3i). On the contrary, the solid line for sulfuric acid demonstrates a pronounced curvature in the regions of both the concentration decrease and its increase (Figure 3i), which is obviously due to the variation in the dissociation degree of the acid as a consequence of the gradual change in pH.

The same effect manifests itself in a different manner for the dependence of the bromate concentration on *E* or on *x* (red lines in Figure 3h,i). This concentration decreases monotonously. Two approximate plots in Figure 3h show a strong difference between them while the solid line for sulfuric acid varies gradually from the starting value which is universal: [BrO_3_^−^] = *c*_tot_ towards the approximate one for α_H_ = 2 because of the proton-concentration diminution. The amplitude of the effect is much weaker for the dependence on *x* (Figure 3i): this plot starts from a universal straight line within an extended range of *x*. For lower values of *x,* the two approximate graphs represent straight lines with a bit smaller slope (independent of the dissociation degree), i.e., parallel to one another. Since this region corresponds to very low proton concentrations, the solid curve for sulfuric acid coincides with the approximate plot for α_H_ = 2 within the whole range of *x* values.

### 2.4. Effect of Added Acid Concentration for c_tot_ = 0.1 M

Figure 4 illustrates how the choice of the concentration of the added sulfuric acid (for the same total concentration of Br atoms, *c*_tot_ = 0.1 M) affects the evolution of various characteristics of the system. This concentration, *c*_A_, varies between 0.3 M where the solute proton concentration, *c*_H_, is ***in excess***, compared to the bromate anion one (*c*_H_ > [BrO_3_^−^] within the whole electrolysis period, see Figure 2), and 0.015 M where the bromate anion is ***in excess*** during the initial stage of the evolution.

Figure 4a,b show that pH varies ***qualitatively*** in the same manner in the course of the electrolysis: a monotonous increase within the initial stage of the process up to a maximal value, then a monotonous decrease to its final value identical to that at the beginning of the electrolysis. However, the shape of these plots fully changes depending on the added acid concentration, c_A_. For its largest value, 0.3 M, the pH variation is relatively weak; it has the form of two (increasing and decreasing) waves, with a plateau between them, in Figure 4a while it is close to two straight lines, crossing at their maxima, in Figure 4b. Lower addition of the acid: *c*_A_ = 0.1 M, increases the amplitudes of the waves and diminishes the width of the plateau in Figure 4a, i.e., to larger slopes and to a curvature of plots in Figure 4b. For even smaller acid additions: *c*_A_ ≤ 0.05 M, the waves and plateau in Figure 4a are replaced by the region of a linearly increasing dependence, followed by a narrow maximum, then a decrease either as a linear function at *c*_A_ = 0.05 M, or drastically at *c*_A_ = 0.03 M or *c*_A_ = 0.015 M. The shapes of the pH variation as a function of *x* = *Q*/*FV c*_tot_ in Figure 4b at *c*_A_ ≤ 0.05 M are more complicated, with a hump for *c*_A_ = 0.05 M, or with a slowly increasing function for *c*_A_ = 0.03 M or *c*_A_ = 0.015 M. The maximal pH value increases strongly within this range of the *c*_A_ values, but even with its lowest value, this pH maximum is only slightly above five, i.e., the solution retains its ***acidic*** character.

The corresponding plots for the solute proton concentration, *c*_H_, at all these additions of sulfuric acid also vary in qualitatively the same manner: a decrease from its initial value (dependent on the amount of added acid) within a narrow potential range, *E*, or linearly with the same slope as a function of *x*; the minimal value of the *c*_H_ concentration diminishes drastically with the decrease in the amount of the added acid (see above on the maximal pH values in Figure 4a,b); then, a return of the *c*_H_ concentration to its initial value.

According to Figure 4c for the dependence of the average oxidation degree of Br atoms, *x* = *Q*/*FV c*_tot_, on the electrode potential, *E*, the variation in the amount of the added sulfuric acid changes radically the shape of these plots. These curves for the sufficiently high acid concentrations, *c*_A_ = 0.3 M and *c*_A_ = 0.1 M, which correspond to an ***excess*** of protons compared to bromate anions (for *c*_A_ = 0.1 M the parallel diminution of the concentrations of both protons and bromate anion with the same rate leads to the ***increase in the total dissociation degree of the acid***, see Section 2.3), are composed of two well-separated waves (transitions: BrO_3_^−^ to Br_2_ and Br_2_ to Br_3_^−^ and Br^−^) with a plateau of *x* between them.

A further diminution of *c*_A_ up to 0.05 M results in practical merging of two waves while an ***inclined plateau*** appears within the medium potential range for two lowest acid concentrations, 0.03 M and 0.015 M. This feature is a consequence of the rapid shift of the bromate-anion reduction wave potential (BrO_3_^−^ to Br_2_) towards less positive potentials if pH increases (Figure 4a), accompanied by a strong deformation of the shape of the wave (curve for *c*_A_ = 0.05 M). For even higher initial values of pH, the BrO_3_^−^-to-Br_2_ transition within the starting stage of the evolution is replaced by the BrO_3_^−^-to-Br^−^/Br_3_^−^ one (see Figure 4e,g,i,k for the variation in the concentrations of Br-containing components, also the discussion below).

There is a ***striking similarity*** in the shapes of the plots for the average oxidation degree of Br atoms, *x* (Figure 4c), and for the bromate-anion concentration, [BrO_3_^−^] (Figure 4d), as function of *E* for the series of the added acid concentrations, except for their small values where [BrO_3_^−^] ***vanishes abruptly*** (more precisely: becoming very small) while the rapid decrease in *x* is suddenly replaced by the universal curve for all values of the added acid concentration (see above).

The reason for such parallelism may be found in Figure 4e for the [BrO_3_^−^] vs. *x* plots, where each line starts from a linear dependence for the initial stage corresponding to Scheme (22) for the BrO_3_^−^ to Br_2_ transition, which is replaced later for two lowest concentrations of the added acid, 0.03 M and 0.015 M, by another straight line with a slightly smaller slope for the BrO_3_-to-Br^−^/Br_3_^−^ transition. This interpretation of the data in Figure 4e is based on the plots for the concentrations of other Br-containing components: Br_2_, Br_3_^−^, and Br^−^ in Figure 4g,i,k, where only Br_2_ is generated from BrO_3_^−^ within the initial range of *x* and the Br_2_ concentration changes according to the same linear line for all amounts of the added acid (Figure 4g) while this concentration changes abruptly its variation to a slow decrease after a transition point (dependent on the amount of added acid), replaced by rapid growth of the Br_3_^−^ and Br^−^ concentrations (Figure 4i,k). Within the final stage of the evolution (after the passage of the second critical point where the bromate concentration becomes very small; see Figure 4d,e) the solution contains mostly the Br_2_, Br_3_^−^, and Br^−^ components so that further transformation between them is independent of pH, which means a universal behavior for this stage in Figure 4c,f–k.

Figure 4f,g for the Br_2_ concentration demonstrate an effect that is of ***key importance for applications of the bromate reduction process in power sources***: the ***maximal Br_2_ concentration*** generated in the course of this process is close to ½ of the initial concentration of bromate, *c*_tot_, e.g., close to 0.05 M for *c*_tot_ = 0.1 M, ***if the initial acidity is sufficiently high***: *c*_A_ ≥ 0.05 M, while ***this maximum of the Br_2_ amount diminishes rapidly for lower c_A_ additions***: *c*_A_ ≤ 0.03 M.

One should also note a complicated variation in the plots for the Br_3_^−^ and Br^−^ concentrations on *E* (Figure 4h,j) or on *x* (Figure 4i,k) as a function of the amount of the added acid. These dependences are given by almost ***universal*** graphs for sufficiently ***high*** *c*_A_ values: *c*_A_ ≥ 0.05 M while one can see a ***strong change in*** these plots for ***lower initial acidities***: *c*_A_ ≤ 0.03 M, where the *x* range of their generation is markedly extended for larger values of *x* (Figure 4i,k). At the same time, the dependence of [Br^−^] on *E* (Figure 4j) is practically ***universal for all values of the added acid concentrations***.

### 2.5. Effect of Initial Bromate Concentration

In order to illustrate the effect of the initial bromate concentration, i.e., of the total concentration of Br atoms in the system, *c*_tot_, Figure 5 presents the graphs analogous to those in Figure 4 but for another value of this concentration: *c*_tot_ = 1 M instead of *c*_tot_ = 0.1 M, while the set of the added acid concentrations, *c*_A_, is the same as that in Figure 4.

It should be noted that after this strong increase in the total Br-atom concentration, ***all*** considered values of *c*_A_ correspond to the case of a ***strong bromate excess*** since the initial concentration of solute protons, *c*_H_^ini^, for the largest acid addition, *c*_A_ = 0.3 M, is close to 0.3 M (since there is practically no dissociation of sulfuric acid along its second step), i.e., it is much lower than *c*_tot_ = 1 M.

According to Scheme (22) for the BrO_3_^−^-to-Br_2_ transition, the solute proton concentration, *c*_H_, is to diminish ***linearly as a function of x*** within the initial *x* range until its value becomes much lower than the initial one, *c*_H_^ini^. Since the slope of these graphs is independent of *c*_H_^ini^, the length of this linear region is shortened for smaller *c*_H_^ini^ values. In terms of the pH vs. *x* dependence (Figure 5b), it means that the initial range of a higher acidity occupies for *c*_A_ = 0.3 M a relatively broad interval: +2 < *x* < +5, with a rapid increase in pH for *x* < +2, while the initial region is practically absent for much lower acid additions, *c*_A_ ≤ 0.05 M, i.e., the region of the drastic increase in pH is located within a narrow interval in the vicinity of *x* = +5.

A similar feature may be noted within the final region of the *x* coordinate (Figure 5b), where the drastic decrease in pH takes place in the close vicinity of the terminal point, *x* = −1, for four acid additions: *c*_A_ ≤ 0.1 M. Only for the largest of the added acid concentrations, *c*_A_ = 0.3 M, rapid decrease in pH starts obviously earlier.

As a consequence, the length of the interval of strongly enhanced pH values within the medium *x* range increases strongly with the diminution of the added acid concentration, up to the almost whole range, from +5 to −1, for the three lowest *c*_A_ values (Figure 5b). Within this region, the pH value increases slowly in the course of the electrolysis, and for all *c*_A_ concentrations, except for 0.3 M, the maximal value of pH belongs to the range: 6 < max pH < 7 for this value of the total Br-atom concentration, *c*_tot_ = 1 M. It means that for the ***quantitative*** analysis, one should take into account the process of the ionic dissociation of water but it may be expected that the ***qualitative*** features of the plots should hardly change significantly.

For the pH dependence on the potential, *E* (Figure 5a), there are regions of sufficiently high and sufficiently low potentials where the pH, i.e., the solute proton concentration, is close to its initial value. Closer to the medium potential range, there are two waves where the *c*_H_ varies between its initial value (dependent on *c*_A_) and a very small one. From the set of plots of *c*_H_ vs. *E* for various acid additions, one can see that they ***merge together*** within this intermediate potential range after passage through the decreasing wave.

This conclusion is confirmed by the dependences of pH vs. *E* (Figure 5a), where this wave occupying a relatively narrow potential range looks like a point of an ***abrupt change in the slope*** between the potential regions where pH is either constant or varies linearly. This linear dependence corresponding to the region of high pH values in Figure 5b is a ***universal*** one for all acid additions, except for the values of the potential where this behavior starts or finishes. The latter corresponds to the maximum pH for both figures (Figure 5a,b) for each *c*_A_ value. A further evolution leads to the diminution of the potential, which is accompanied by an extremely rapid decrease in pH. For example, a detailed analysis of the plot for *c*_A_ = 0.3 M has shown that ***the pH change by several units takes place within a potential region of a fraction of 0.1 mV wide***.

Comparison of the plots for pH vs. *E* or *x* for two different values of the total Br-atom concentrations, *c*_tot_ = 0.1 M (Figure 4a,b) and *c*_tot_ = 1 M (Figure 5a,b), reveals a close similarity of the former graphs (red and black lines) for two lowest acid additions with the latter ones (magenta and green lines) for two highest acid additions. One may note that the values of the ratio, *c*_A_/*c*_H_, for these curves are equal to 0.3 and 0.15 for the former case and 0.3 and 0.1 for the latter, i.e., one might assume that it is this ratio of the initial concentrations which determines the ***qualitative shapes*** of plots in Figure 4a,b and Figure 5a,b. This similarity can also be traced for the other graphs in Figure 4 and Figure 5.

The shapes of the plots in Figure 5c–k for the ***highest*** acid concentrations, *c*_A_ = 0.3 M and *c*_A_ = 0.1 M, also resemble those of the graphs of Figure 4c–k for the ***lowest*** concentrations, *c*_A_ = 0.03 M and *c*_A_ = 0.015 M, taking into account that all graphs for Br-containing components should be normalized by the total concentration of Br atoms in the system. Shapes of the curves in Figure 5c–k for ***lower*** acid concentrations, *c*_A_ ≤ 0.05 M, continue the same tendencies found for those at *c*_A_ = 0.1 M.

In particular, the ***diminution of the acid addition reduces progressively the amount of solute bromine***, Br_2_ (Figure 5f,g). The maximal Br_2_ concentration at *c*_A_ = 0.3 M exceeds 0.25 M; it means that within the range of *E* or *x* where [Br_2_] is over its saturation limit (shown by a horizontal black line at 0.185 M) its calculated values ***overestimate*** the equilibrium ones since the excess of solute bromine is to give the ***liquid bromine phase*** while the solute bromine concentration rests at its saturation value, with recalculations of the concentrations of the other Br-containing components [11,12]. For all other values of the acid additions: *c*_A_ ≤ 0.1 M, the maximal values of the solute bromine concentration are already much lower than the saturation one. Thus, ***a proper choice of acid concentration allows one to avoid the formation of the liquid bromine phase***. This conclusion is of ***primary importance for the application of the bromate reduction process in power sources*** since this phase has various harmful effects on their functioning.

Similar to the tendency already observed in Figure 4h,i for the two ***lowest*** acid additions, *c*_A_ = 0.03 M and *c*_A_ = 0.015 M, the decrease in *c*_A_ also leads to the diminution of the maximal Br_3_^−^ concentration in Figure 5h,i, this time already within the whole set of acid concentrations. As for the region of its accumulation, it is extended for lower *c*_A_ values vs. the *x* coordinate (Figure 5i) while an opposite tendency is observed vs. the *E* coordinate. For the lowest acid concentrations, *c*_A_ ≤ 0.05, this region occupies almost fully the whole *x* range, starting already in the vicinity of *x* = +5.

An analogous extension vs. the *x* coordinate is also visible for the Br-concentration, which increases in an almost linear way from 0 around *x* = +5 to *c*_tot_ at *x* = −1.

Thus, for sufficiently small acid additions, the bromate reduction gives immediately the final product, bromide anion, with the generation of some amounts of solute bromine and of tribromide anion, dependent on the added acid concentration.

## 3. Materials and Methods

The set of equations has been solved numerically with the use of the Mathcad software. For each value of the potential, *E*, the starting value of pH is equal to that found for the previous *E* value. Then, the equation for the bromide concentration is solved numerically. Thus found values of all Br-containing concentrations are inserted into Equation (18) to solve it numerically. The result of this step for pH is used again to solve the equation for [Br^−^], etc., until the change in the values of pH and [Br^−^] becomes sufficiently small.

## 4. Conclusions

The analysis performed in this study has confirmed the previous results [11,12] based on the constant-pH approximation that for a sufficiently high acid content (compared to the bromate one), the maximal concentration of molecular bromine within the medium stage of the bromate catholyte electrolysis reaches ***half of the initial bromate concentration***, i.e., its very high value for a large initial bromate concentration. On the contrary, the ***diminution of the starting amount of the acid***, for a fixed initial bromate concentration, leads to a ***marked decrease in the maximal amount of the molecular bromine***.

For sulfuric acid used as an acidifier and for the initial bromate concentration of 0.1 M, the above effect of the suppression of Br_2_ generation becomes noticeable when the initial content of H_2_SO_4_ is below 0.03 M.

A higher initial concentration of bromate, 1 M, with the addition of 0.3 M sulfuric acid, results in the calculated maximal concentration of bromine at the level of 0.25 M, which exceeds its solubility so that it creates a danger of liquid bromine formation within the medium interval of the bromate-to-bromide conversion. However, for a lower value of the initial acid concentration: c_A_ < 0.1 M, and the same initial bromate concentration, 1 M, the maximal values of the solute bromine concentration are already much lower than its solubility one. Thus, a proper choice of the added acid concentration allows one ***to avoid the appearance of the liquid bromine phase***. It implies that under such conditions ***the bromate reduction gives immediately the final product, bromide***, without a significant accumulation of intermediate species, Br_2_, Br_3_^−^, etc.

One should note that the developed algorithm can be applied almost without changes to the analysis of the inverse process, i.e., the electrooxidation of bromide inside MEA with a proton-exchange membrane. This allows one to generalize the proposed approach, based on the combination of the hydrogen-ion balance with the thermodynamics of bromine compounds, in order to predict the evolution of the composition of electrolyzed media in such applications as the electrosynthesis of bromates, the production of disinfectant solutions by electrolysis of bromides, the removal of bromates after water treatment by ozonation, and the electrolytic production of hydrogen from seawater.

## Figures and Tables

**Figure 1 ijms-24-15297-f001:**
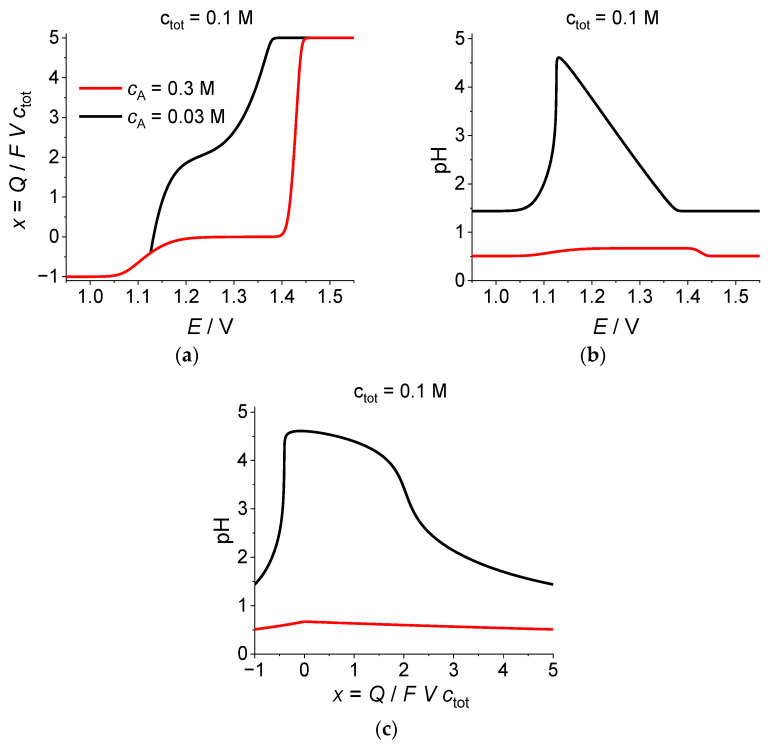
Evolution of (**a**) the average oxidation degree of Br atoms, *x* = *Q*/*FV c*_tot_, or (**b**) pH as functions of the electrode potential, *E*, or (**c**) pH as a function of the average oxidation degree of Br atoms, *x*. Parameters: *c*_tot_ = 0.1 M, *c*_A_ = 0.3 M (red lines) or 0.03 M (black lines).

**Figure 2 ijms-24-15297-f002:**
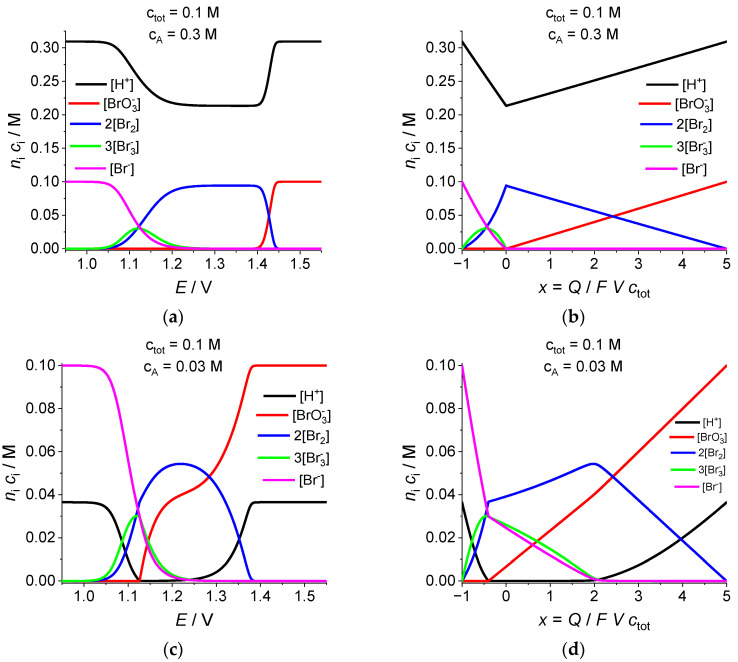
Evolution of the concentrations of the solute components (in terms of the Br-atom concentration inside each component, *n*_i_ *c*_i_, for 1 ≤ i ≤ 7 in Table 1) as a function of (**a**,**c**) the electrode potential, *E*, or (**b**,**d**) the average oxidation degree of Br atoms, *x* = *Q*/*FV c*_tot_. Attribution of a color of line to a solute component is specified in the legend. Lines for Br_5_^−^, HBrO, and BrO^−^ are omitted because of low values of their concentrations. Parameters: *c*_tot_ = 0.1 M, *c*_A_ = 0.3 M (**a**,**b**) or 0.03 M (**c**,**d**).

**Figure 3 ijms-24-15297-f003:**
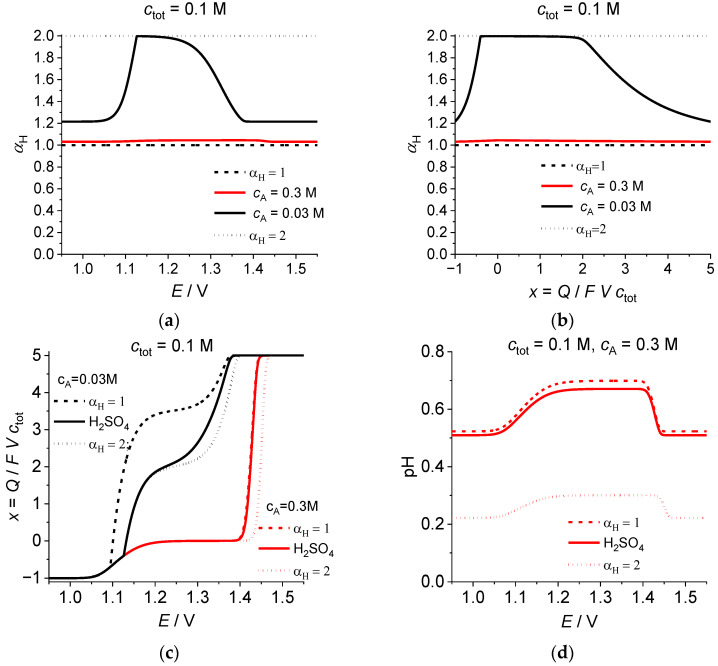
Dependence of (**a**,**b**) the total dissociation degree of sulfuric acid, α_H_, Equation (20), or (**c**) the average oxidation degree of Br atoms, *x* = *Q*/*FV c*_tot_, or (**d**–**g**) pH, or (**h**,**i**) *c*_H_ and [BrO_3_^−^] on (**a**,**c**,**d**,**f**,**h**) the electrode potential, *E*, or (**b**,**e**,**g**,**i**) the average oxidation degree of Br atoms, *x* = *Q*/*FVc*_tot_. Solid lines in (**c**), or (**d**,**f**,**h**) or (**e**,**g**,**i**) are identical to those in Figure 1a, or in Figure 1b and Figure 2c, or in Figure 1c and Figure 2d, respectively. Dotted and dashed lines are calculated for the complete dissociation of the second stage, α_H_ = 2, and for no dissociation of this stage, α_H_ = 1, respectively. Parameters: *c*_tot_ = 0.1 M, *c*_A_ = 0.3 M, or *c*_A_ = 0.03 M (indicated inside each graph).

**Figure 4 ijms-24-15297-f004:**
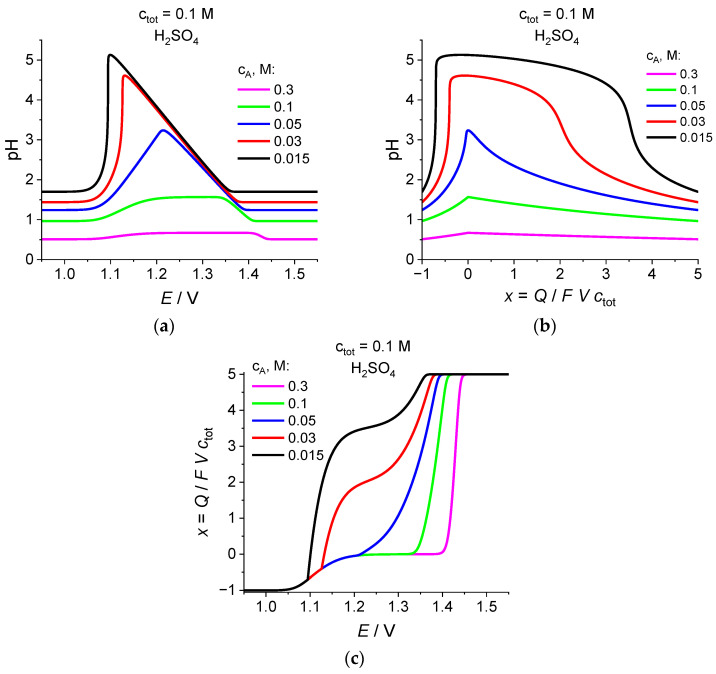
Dependence of (**a**,**b**) pH, or (**c**) the average oxidation degree of Br atoms, *x* = *Q*/*FV c*_tot_, or (**d**,**e**) the bromate-anion concentration, [BrO_3_^−^], or (**f**,**g**) the solute bromine concentration, [Br_2_], or (**h**,**i**) the tribromide-anion concentration, [Br_3_^−^], or (**j**,**k**) the bromide-anion concentration, [Br^−^], on (**a**,**c**,**d**,**f**,**h**,**j**) the electrode potential, *E*, or (**b**,**e**,**g**,**i**,**k**) the average oxidation degree of Br atoms, *x* = *Q*/*FVc*_tot_, for various values of the added acid concentration, *c*_A_ (these values are indicated inside each graph), while the total number of Br atoms, i.e., the initial bromate concentration is equal to *c*_tot_ = 0.1 M. Lines for *c*_A_ = 0.3 M and *c*_A_ = 0.03 M are identical to the corresponding plots in Figure 1 and Figure 2.

**Figure 5 ijms-24-15297-f005:**
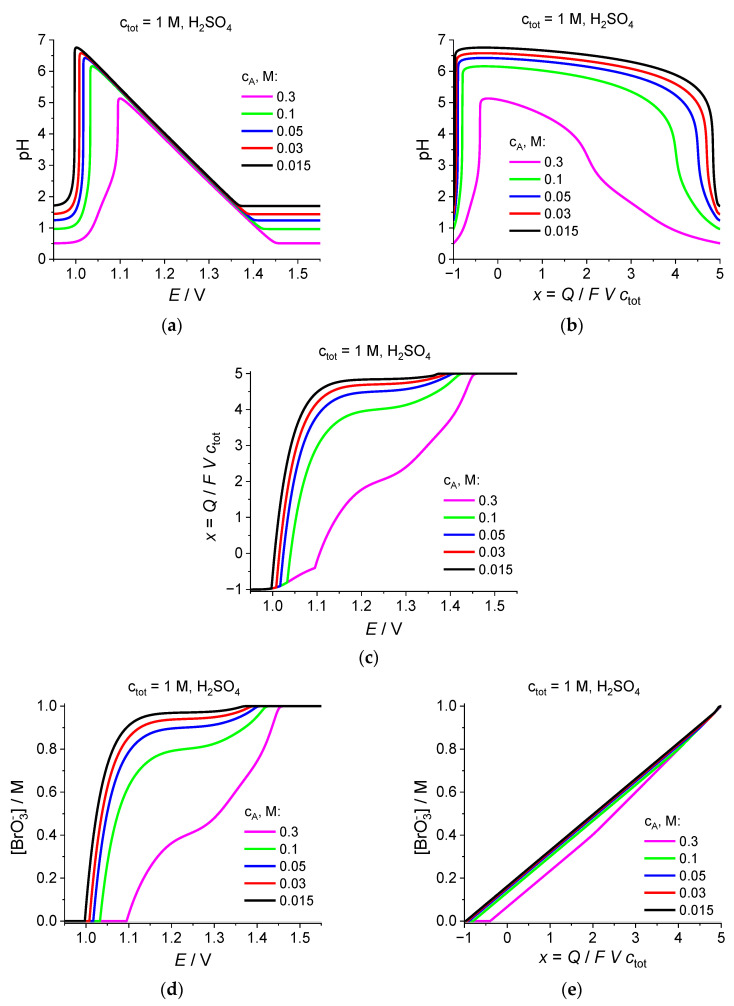
See Figure 4, *c*_tot_ = 1 M. Dependence of (**a**,**b**) pH, or (**c**) the average oxidation degree of Br atoms, *x* = *Q*/*FV c*_tot_, or (**d**,**e**) the bromate-anion concentration, [BrO_3_^−^], or (**f**,**g**) the solute bromine concentration, [Br_2_], or (**h**,**i**) the tribromide-anion concentration, [Br_3_^−^], or (**j**,**k**) the bromide-anion concentration, [Br^−^], on (**a**,**c**,**d**,**f**,**h**,**j**) the electrode potential, *E*, or (**b**,**e**,**g**,**i**,**k**) the average oxidation degree of Br atoms, *x* = *Q*/*FV c*_tot_, for various values of the added acid concentration, *c*_A_ (these values are indicated inside each graph), while the total number of Br atoms, i.e., the initial bromate concentration, is equal to *c*_tot_ = 1 M.

**Table 1 ijms-24-15297-t001:** Total oxidation number (*x*_i_) of Br atom(s) as well as numbers of Br (*n*_i_), H (*h*_i_), and O (*o*_i_) atoms inside one Br-containing species of type i.

i	1	2	3	4	5	6	7	8	9
	Br^−^	Br_3_^−^	Br_5_^−^	Br_2_	HBrO	BrO^−^	BrO_3_^−^	Br_2_^liq^	Br_2_^vap^
*n* _i_	1	3	5	2	1	1	1	2	2
*x* _i_	−1	−1	−1	0	+1	+1	+5	0	0
*h* _i_	0	0	0	0	1	0	0	0	0
*o* _i_	0	0	0	0	1	1	3	0	0

## Data Availability

Data is contained within the article.

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
