# Peer review of "Evolution of the Bromate Electrolyte Composition in the Course of Its Electroreduction inside a Membrane–Electrode Assembly with a Proton-Exchange Membrane"

_ijms, 2023, doi:10.3390/ijms242015297_

Round 1

Reviewer 1 Report

Word file

Check English Language

Author Response

Thank you very much for your comments on our article! We have made an effort to respond to each of them in great detail.

Reviewer 2 Report

THe paper is well writtena and sound. I reccommend some minor revisions before publication:

1) revise english in the introduction, there are at eleast a couple of sentences whihc need to be rephrased

2) add the meaning of n, x, h and o in the caption of Table 1, at least their names

3) define F in Eq.8

4) Page 5: NH0 + NHA0, is equal to 2 V cA. please explain why

5) Figure 1b: the label of y axis is wrong

6) all figures would be clearer if the different panels would be placed close one to the other in landscape orientation (not portrait)

7) page 9: orange and magenta lines in Figures 2a,b; actually there are no orange lines in this figure

8) page 11: thie behavior may be interpreted via Scheme (21) where 5 of 6 H+; there is no scheme (21)

9) Fig. 3a: differentiate between lines with alpha_H =1 or 2 (for example use different colors or dashes)

Some changes in the introduction are nedeed

Author Response

(The authors gave the same response as above.)
